# Simultaneous Estimation of Gender Male and Atrial Fibrillation as Risk Factors for Adverse Outcomes Following Transcatheter Aortic Valve Implantation

**DOI:** 10.3390/jcm9123963

**Published:** 2020-12-07

**Authors:** Yuichi Chikata, Hiroshi Iwata, Shinichiro Doi, Takehiro Funamizu, Shinya Okazaki, Shizuyuki Dohi, Ryosuke Higuchi, Mike Saji, Itaru Takamisawa, Harutoshi Tamura, Atsushi Amano, Hiroyuki Daida, Tohru Minamino

**Affiliations:** 1Department of Cardiovascular Biology and Medicine, Juntendo University Graduate School of Medicine, Tokyo 113-8421, Japan; ychikata@juntendo.ac.jp (Y.C.); doies@juntendo.ac.jp (S.D.); t-funamizu@juntendo.ac.jp (T.F.); shinya@juntendo.ac.jp (S.O.); daida@juntendo.ac.jp (H.D.); t.minamino@juntendo.ac.jp (T.M.); 2Department of Cardiovascular Surgery, Juntendo University Graduate School of Medicine, Tokyo 113-8421, Japan; shiz-d@juntendo.ac.jp (S.D.); a-amano@juntendo.ac.jp (A.A.); 3Department of Cardiology, Sakakibara Heart Institute, Tokyo 183-0003, Japan; rhiguchi@shi.heart.or.jp (R.H.); msaji@shi.heart.or.jp (M.S.); itakami@shi.heart.or.jp (I.T.); 4Department of Cardiology, Pulmonology and Nephrology, Yamagata University School of Medicine, Yamagata 990-9585, Japan; htamura@med.id.yamagata-u.ac.jp

**Keywords:** TAVI, long-term outcome, AF, gender difference

## Abstract

Accurate outcome prediction following transcatheter aortic valve implantation (TAVI) has gained further importance along with expanding its indication to patients with a lower surgical risk. Although previous studies have evaluated the prognostic impacts of gender and atrial fibrillation (AF) in TAVI patients, these two factors have rarely been addressed simultaneously. This retrospective observational study based on a multicenter TAVI registry involved 1088 patients who underwent TAVI between May, 2010 and February, 2020 at 3 hospitals in Japan. Participants were divided into 4 groups by gender and pre-existing AF, such as Female AF (−) (*n* = 559), Male AF (−) (*n* = 266), Female AF (+) (*n* = 187) and Male AF (+) (*n* = 76). Primary and secondary endpoints were death due to any and cardiovascular cause, and the composite of all-cause death and heart failure hospitalization, respectively. The median follow-up period was 538 days. Cumulative incidences of primary and secondary endpoints were lower in the Female AF (−) group compared to the other 3 groups. Adjusted multivariate Cox proportional hazard analyses showed an independent association of either or both of male gender and AF with adverse outcomes, when compared to the group with none of these (hazard ratios and 95% confidence intervals vs. Female AF (−) (reference) for all-cause death of Male AF (−): 2.7, 1.6–4.6, *p* < 0.001, Female AF (+): 3.5, 2.1–6.0, *p* < 0.001, and Male AF (+): 3.9, 1.9–7.8, *p* < 0.001), while there was no evidence of their synergistic prognostic impact. Male gender and being complicated by AF independently, but not synergistically, predicted poor long-term outcomes in patients undergoing TAVI.

## 1. Introduction

Transcatheter aortic valve implantation (TAVI) has emerged as an alternative treatment for patients with severe aortic stenosis (AS) who are at high-risk or inoperable for surgical aortic valve replacement (SAVR) [1,2]. More recently, accumulating evidence has indicated the noninferiority of TAVI compared to SAVR in intermediate or low surgical risk individuals with severe AS [3,4,5,6]. Significant advancements in valve platforms, operator experience, and technical refinements in the TAVI procedure have substantially improved its safety and short-term outcomes and the indication of TAVI has been expanding for the entire spectrum of symptomatic severe AS [5,6]. Therefore, accurate risk stratification for predicting longer-term outcomes following TAVI is of further clinical importance in order to develop treatment strategies with appropriate management in patients with severe AS. Although several risk scores specifically for TAVI have been recently developed [7], conventional risk scores, such as the logistic European System for Cardiac Operative Risk Evaluation (EuroSCORE) [8], EuroSCORE II [9] and STS-PROM [10], which were designed for estimating short-term or procedural risk in cardiac surgery, are used in the majority of patients.

From amongst a wide range of prognostic factors in TAVI patients, gender difference and atrial fibrillation (AF) have been intensively evaluated. The impact of gender difference on outcomes has been a topic of clinical interest since the introduction of TAVI into clinical practice [11]. AF is a major risk factor which associates with mortality and morbidities not only in the entire population [12], but also in those with AS [13] and in individuals who underwent TAVI [13]. However, despite a large body of evidence indicating a significant gender difference in AF [14], no study has yet to evaluate the impact of these two factors simultaneously on outcomes after TAVI. This retrospective cohort study tested whether the prognostic impact of gender and AF is synergistic in patients after TAVI in order to obtain accurate risk estimation in patients undergoing TAVI.

## 2. Materials and Methods

### 2.1. Multi-Center TAVI Registry Database

This study is a retrospective analysis of a prospective multi-center registry database of patients who underwent TAVI at three institutions in Japan; Sakakibara Heart Institute, Juntendo University Hospital and Yamagata University Hospital. This study was performed in accordance with the Declaration of Helsinki and with approval from the Institutional Review Board (IRB) of Sakakibara Heart Institute (IRB-ID: 17-048), Juntendo University (IRB-ID: 17-263) and Yamagata University (IRB-ID: 2019-407), respectively and this registry is publicly registered in the University Medical Information Network Japan—Clinical Trials Registry, (UMIN000031133). Written informed consent was obtained from all participants for this registry.

### 2.2. Participants, Definition of Endpoints and Follow-Up Period

This study enrolled 1088 patients who underwent TAVI between May 17, 2010 and February 27, 2020. Participants were divided into four groups by gender, and without or with pre-existing AF; females without AF (Female AF (−)) (*n* = 559, 51.4%), males without AF (Male AF (−)) (*n* = 266, 24.4%), females with AF (Female AF (+)) (*n* = 187, 17.2%), and males with AF (Male AF (+)) (*n* = 76, 7.0%), and the incidence and risk of subsequent endpoints following TAVI were assessed. In the present study, AF was defined as atrial fibrillation or atrial flutter which was documented by any ECG prior TAVI whether or not with antiarrhythmic medications and anticoagulants. The primary endpoint was defined as any cause mortality, as well as cardiovascular (CV)-related death following TAVI. The secondary endpoint was the composite of all-cause death and heart failure (HF) hospitalization after discharge from the hospital. The follow up period was a maximum of 5 years since the TAVI procedure and the median follow up period was 538 days.

### 2.3. Statistical Analysis

Quantitative variables are presented as the mean ± standard deviation or median with interquartile range (IQR) in accordance with the results of the Shapiro-Wilk normality test. Categorical variables are presented as the numbers and percentages. Quantitative data across groups were compared using the ANOVA test or the Kruskal-Wallis test. Unadjusted Kaplan-Meier analysis evaluated the time to the cumulative incidence of endpoints followed by the log-rank test for comparisons. Univariate and multivariate Cox proportional hazards regression analyses catabolically calculated hazard ratios (HRs) of either or both of being male and with AF, including groups of Male AF (−), Female AF (+) and Male AF (+) by using the Female AF (−) group as a reference (HR:1). Covariates used in the models of multivariate analysis for all-cause mortality (Model 1) and for all-cause mortality and CV mortality and the composite of all-cause death and HF hospitalization (Model 2) were selected based on background demographics and findings in univariate analyses. Model 1 included age (a continuous variable), TAVI procedure later than 2017, body mass index (BMI) (a continuous variable), New York Heart Association (NYHA) HF classification, diabetes, chronic obstructive pulmonary disease (COPD), peripheral artery disease (PAD), logistic EURO score (a continuous variable), hemoglobin (a continuous variable), renal function (estimated glomerular filtration rate (eGFR), a continuous variable), moderate or severe mitral and tricuspid regurgitation (more than moderate MR and TR), implanted valve size, and low-flow low-gradient (LF-LG) AS in addition to categorical analysis of the 4 groups. Model 2 included age, NYHA HF classification, history of HF, and renal function (eGFR, a continuous variable). Statistical significance was defined as a *p*-value < 0.05 and analyses were performed using statistical software (JMP Pro 12.0; SAS Institute Inc., Cary, NC, USA and IBM SPSS Statistics, Version 24.0. Armonk, NY, USA).

## 3. Results

### 3.1. Baseline Demographics, Medications, Procedural Characteristics and Incidence of Complications of 4 Groups Divided by Gender and AF

The background demographics, comorbidities, medications, procedural characteristics, and devices among the groups are listed and compared in Table 1. Patients in the Male AF (−) group were significantly younger compared to the Female AF (−) and Female AF (+) groups. There was a significant gender difference in comorbidities, with diseases such as diabetes, history of stroke, COPD and history of coronary revascularization being higher in males than females. In contrast, among patients with AF, the ratios of individuals with a history of stroke and HF, and chronic kidney disease (CKD) were significantly higher than those without AF. The prevalence of past HF was almost double in patients with AF. Consistently, NT-proBNP and the NYHA HF classification were higher, while eGFR was lower in AF patients. Among the four groups, the ratio of stroke history decreased in the following order; Male AF (+), Female AF (+), Male AF (−) and Female AF (−). Conversely, left ventricular ejection fraction (LVEF) was the lowest in the Male AF (+) group and the highest in the Female AF (−) group. Interestingly, both the preprocedural peak and mean pressure gradient through the aortic valve were in the same order as EF, indicating a higher incidence of low gradient AS in the Male AF (+) group. TAVI procedures and devices, such as time duration, contrast media, approach, type of anesthesia and type of transcatheter heart valve (THV) (balloon- vs. self-expandable THV) were similar among the groups.

### 3.2. In-Hospital and Long-Term Outcomes Following TAVI in the 4 Patient Groups Divided According to Gender and AF

During the 5 -year follow-up period since the TAVI procedure, the incidences of identified all-cause mortality, CV mortality, and the composite of all-cause death and HF hospitalization were 141 (13.0%), 59 (5.4%), and 183 (16.8%) out of 1088 participants, respectively. The incidences of most in-hospital adverse outcomes other than acute kidney injury were similar among the groups (Appendix A). Meanwhile, the crude long-term event rates for primary and secondary endpoints represented by the number of events per 1000 person years were significantly different among the groups. The number of all endpoints were highest in the Male AF (+) group and constantly the lowest in the Female AF (−) group, while the Male AF (−) and Female AF (+) groups were in between (Figure 1). Similarly, unadjusted Kaplan-Meier analysis followed by the log-rank comparison test showed that the cumulative incidence of all endpoints was constantly lower in the Female AF (−) group compared to the other three groups (Figure 2a and Appendix A). It is interesting to point out that the significantly higher cumulative all-cause and CV mortality rates in patients with AF compared to those without it were present only in female patients, while the incidence of the composite of all-cause death and HF hospitalization was significantly higher in AF patients in both genders (Figure 2b).

### 3.3. Simultaneous Risk Assessment of Gender Male and AF Following TAVI for All-Cause and CV Mortalities and the Composite of All-Cause Death with HF Hospitalization

Based on the findings of background demographics (Table 1) and univariate analyses (Appendix A), the prognostic impact of the gender difference and AF was categorically assessed by multivariate Cox hazard analysis by using the Female AF (−) group as a reference (HR:1.0). Two models were used for multivariate analysis. In the analysis using Model1, being male and/or complicated by AF were associated with a significantly increased risk for all-cause death compared to the Female AF (−) group (Figure 3a and Appendix A). Moreover, multivariate analyses using Model2 with fewer covariates were performed for simultaneous evaluation of these factors not only for all-cause death, but also CV death the composite of all-cause death and HF hospitalization and sole heart failure hospitalization (Figure 3b and Appendix A). For death from any cause and CV cause, the HRs for either being male and/or having AF, such as Male AF (−), Female AF (+) and Male AF (+) were significantly higher than those with neither (Female AF (−)) for all endpoints. Moreover, the risks for patients in the Male AF (+) group of all-cause and CV mortalities were more than three times (hazard ratios: 3.4 and 4.4, respectively) compared with Female AF (−) group. For HF hospitalization, the risk of AF was more evident compared to any gender difference.

## 4. Discussions

This study simultaneously evaluated the impact of gender and preprocedural AF on long-term outcomes following TAVI. The occurrence ratio and risk of all-cause and CV mortalities and the composite of all-cause death and HF hospitalization were stringently assessed in the four patient groups categorized by gender and AF, which avoided the statistical interaction between AF and gender in this population. The primary findings are as follows, (1) The incidence rates of in-hospital mortality and critical events were similar among the four groups. However, the crude numbers of incidences of all three endpoints within 5 years of TAVI were significantly different among the groups; it was lowest in the Female AF (−) group and the highest in the Male AF (+) group. (2) Kaplan-Meier analyses constantly showed the lowest cumulative incidences of all endpoints in the Female AF (−) group, and this was significantly lower than in the other three groups. (3) Moreover, multivariate Cox proportional hazard analyses using two models showed a significantly increased risk of all endpoints by either or both being male and having AF. However, there was no evidence of synergistically elevated risk by having both of these two factors, while they were independently associated with adverse outcomes after TAVI.

Aortic valve replacement reduces cardiac afterload and significantly improves hemodynamics in patients with AS, and substantial technical refinement in TAVI has reduced the risk of its perioperative risk over time [15]. However, there remains a significant residual risk and the long-term mortality rate following TAVI is still high [16]. Accordingly, an accurate risk prediction in candidate patients for TAVI not only in terms of periprocedural, but also longer-term outcomes has become more important, as it has been firmly established as a therapeutic strategy for severe AS with the rapid expansion of its indication for individuals with lower surgical risk [5]. However, a reliable risk stratification method which can be applicable to a wide range of patients undergoing TAVI has yet to be established. Therefore, several conventional risk scores have been widely used for estimating short-term risk following TAVI, and for deciding upon treatment strategies in a majority of patients with AS. However, they were developed for risk prediction in cardiac surgery, and might have limitations in their application for predicting longer-term outcomes in TAVI. Therefore, a more appropriate model specifically for TAVI is needed [17,18]. In a previous analysis of the registry database of the present study, the usefulness of the STS-ACC TAVR risk score [7], which was developed specifically for TAVI patients, was validated [19]. In that study by Saji et al., the reliability of the STS-ACC TAVR score in predicting all-cause mortality was higher than the conventional STS-PROM, EuroSCORE and EuroSCORE II. More interestingly, the risk prediction accuracy of the STS-ACC TAVR score is further increased when used in conjunction with AF in addition to serum level of albumin and BMI. These findings suggest AF is valuable for risk prediction after TAVI. Accordingly, the modified STS-ACC TAVR score by adding AF, serum albumin and BMI might be further useful for accurate prognostic prediction in patients following TAVI.

The evidence regarding the impact of gender difference on the short- and long-term outcome following TAVI has been extensively evaluated, but it is still inconsistent. Some studies have reported superiority in mid- to long-term survival [20,21], while similar or worse survival in women compared to men was reported in other studies [22,23]. A large-scale cohort study using the STS-ACC registry and a meta-analysis of five major TAVI trials have demonstrated an increased incidence of vascular complications and an increased in-hospital or short-term (30-day) mortality rate after TAVI in women [24,25]. However, the present study showed similar short-term, but significantly better longer-term outcomes in women than men. The poorer short-term prognosis in previous studies in women has been considered to be associated with a significantly higher rate of non-transfemoral access TAVI (non-TF TAVI) in women compared to men [26,27]. Therefore, the very limited number of non-TF TAVI (8.7%) in this study is in accordance with the global trend over time [28,29], and similar ratios of non-TF TAVI between genders may be associated with no difference in short-term outcomes after TAVI in women and men. Moreover, the ratios of approach sites of TAVI among the four groups regarding gender and AF were similar in the present study. These findings indicate that the effect of the TAVI procedure on outcomes was limited in this study.

Pressure overload in the left atrium is induced by AS [30] with the promotion of pathological atrial remodeling and AF leading to increased risk of mortalities and morbidities, such as stroke, and HF [26]. A growing body of evidence is confirming the adverse impact of pre-existing and new-onset AF on outcome in patients after TAVI [27,31]. In this study, the overall incidence of pre-existing AF was 24.3%, which was consistent with those in previous observations, which ranged from 16% to 51.1% [13]. However, it was similar in both genders in this study, which was not consistent with previous studies demonstrating a higher prevalence of AF in male patients who underwent TAVI [25]. Nevertheless, it is interesting to note the impact of AF on all-cause and CV mortality was significant in woman and not in men. Although the substantially smaller sample size in men may have resulted in an underpowered analysis in this study, it is possible that there might be a gender difference in the prognostic impact of AF following TAVI. Given the gender differences in the epidemiology, clinical presentations and prognosis of AF [32], any additional or synergistic association of these two factors for adverse outcomes after TAVI can be postulated. A number of studies separately evaluated the prognostic effect of a gender difference and that of AF on short- and long-term outcomes following TAVI. However, the impacts of these two have been rarely assessed simultaneously. The findings in the present study suggest that being male and AF were independently and significantly related to an increased risk of all-cause and CV mortality and the composite of all-cause death and HF hospitalization after TAVI. However, the presence of both factors did not significantly increase the risk over that of being male and having AF alone, although the HRs were continuously highest for all endpoints with both of them. Therefore, these findings suggest no synergistic, but a slightly additive prognostic effect between being male and AF for adverse outcomes after TAVI.

## 5. Limitations of the Study

This study needs to be interpreted in light of the following limitations. First, the number of institutions and patients that participated in this TAVI registry is relatively limited, and the retrospective nature may cause unaccounted confounding factors, which were not recorded or were not even included in the model, may mediate and lead to the outcomes. Second, the temporal change in prognostic impacts of gender and AF in patients who underwent TAVI also have to be taken into consideration. Recently, Itzhaki et al. demonstrated the attenuation of survival superiority of female patients in “contemporary” (2013–2016) TAVI patients compared to “earlier” (2008–2012) patients [33]. In the present study, although the vast majority (96.3%) of participants were classified as “contemporary” and the model used in multivariate analysis included a variable regarding the year of TAVI (before and after 1 January 2017) and still found significance in the elevated risk of gender male and AF, there remains a temporal difference in the prognostic impact of gender and AF on patients following TAVI. Third, the potential synergistic impact of being male and having AF after TAVI in any specific subpopulation cannot be completely excluded in this study. Fourth, similar to another large-scale Japanese TAVI registry [34], the ratio of female patients is substantially higher in this study, compared to those of other regions [7,33,35]. The relatively lower number and ratio of male patients than other registries might have had an effect on the present findings. Therefore, the findings of this study may need to be confirmed in a future global study by integrating data from real-world registries worldwide.

## 6. Conclusions

This retrospective cohort analysis of a registry database in Japan demonstrated a significant and independent association of being male and preprocedural AF with increased risk of critical CV endpoints. While we found no evidence of a synergistic prognostic effect between these two factors for adverse outcomes, they were independently associated with adverse outcomes after TAVI. The present findings indicate the clinical significance of gender and AF for identifying better treatment strategies and the need for careful postprocedural management in patients following TAVI.

## Figures and Tables

**Figure 1 jcm-09-03963-f001:**
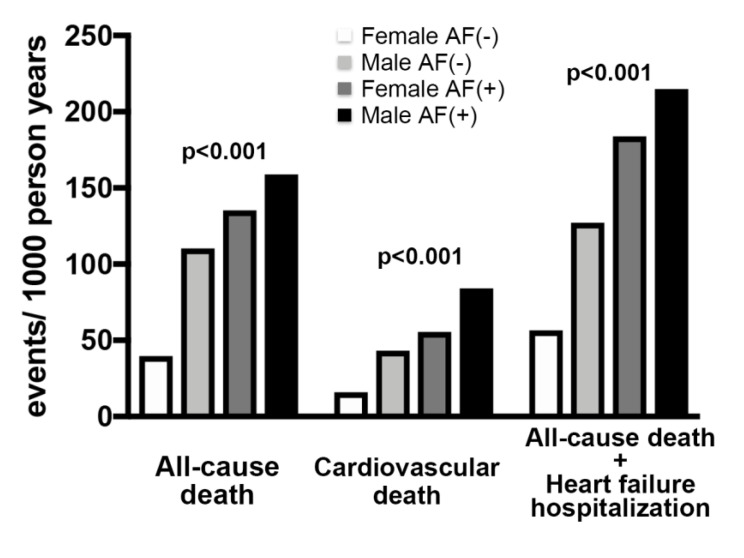
Crude numbers of adverse events following TAVI in the 4 study groups divided according to gender and preprocedural AF. The number per 1000 person years of all-cause and cardiovascular mortalities, and the composite of all-cause death and heart failure hospitalization after TAVI were significantly different among the groups. The order of all events was identical; from the lowest to the highest; Female AF (−), Male AF (−), Female AF (+) and Male AF (+).

**Figure 2 jcm-09-03963-f002:**
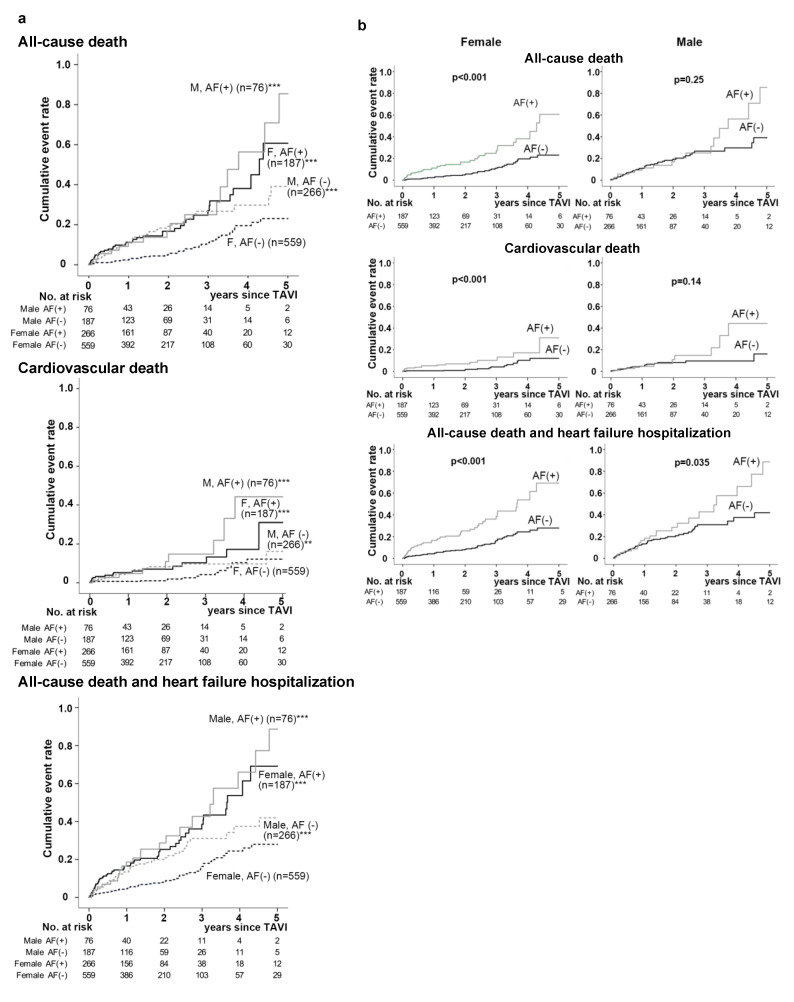
Cumulative incidence of adverse outcomes following TAVI in the 4 study groups according to gender and AF (**a**), and those in patients with and without AF, female and male (**b**). (**a**) cumulative incidence of all-cause and cardiovascular mortality and the composite of all-cause mortality with heart failure hospitalization in the Female AF (−), Male AF (−), Female AF (+) and Male AF (+) groups. ** and *** indicate *p* < 0.001 and < 0.0001 in log-rank test of Kaplan-Meier curves compared to that of the Female AF (−) group. (**b**) Kaplan-Meier curves of patients with and without AF (AF (+) and AF (-)) are separately drawn for females and males.

**Figure 3 jcm-09-03963-f003:**
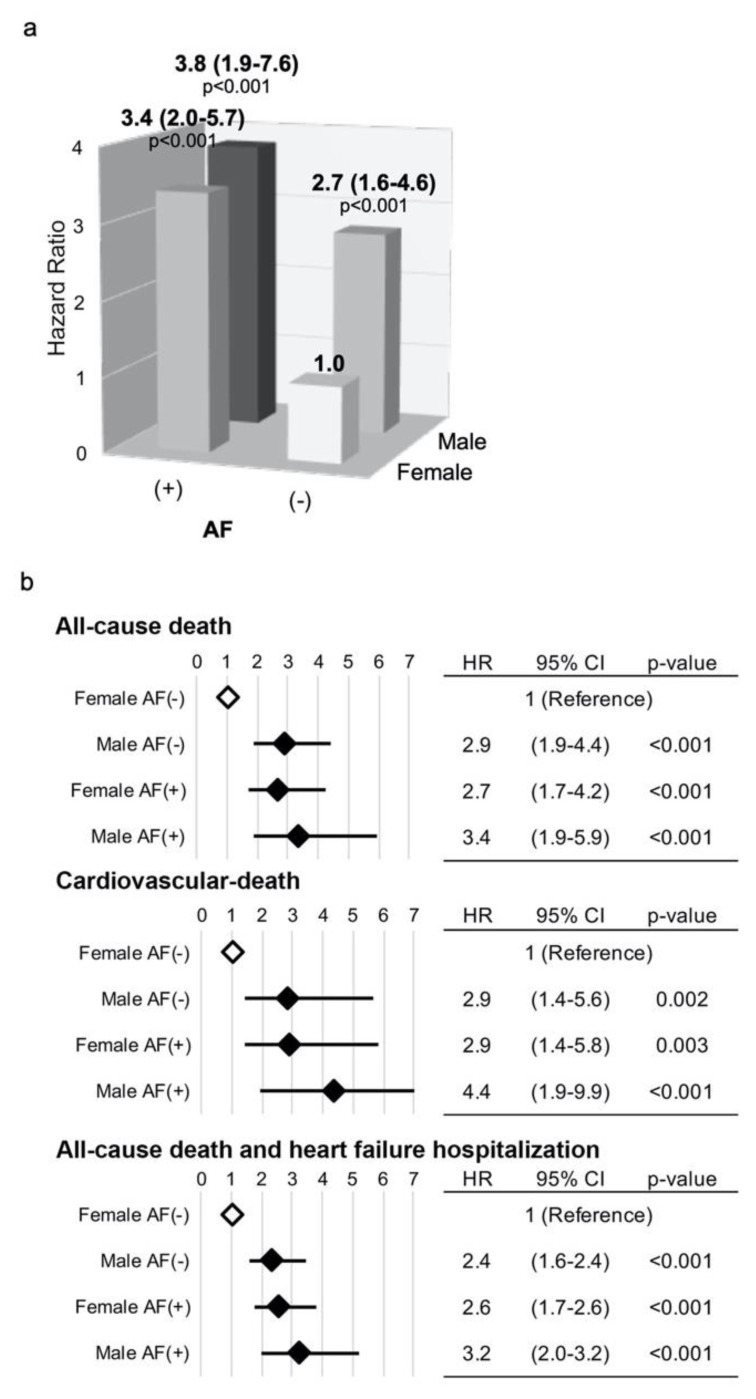
Hazard ratios for all-cause mortality in the 4 patient groups using Model 1 (**a**) and those for all-cause and cardiovascular mortalities and the composite of all-cause mortality and heart failure hospitalization using Model 2 (**b**). Hazard ratios, 95% confidence intervals (CI) and p-values for all-cause mortality following TAVI in the Female AF (−), Male AF (−), Female AF (+) and Male AF (+) groups calculated by Cox proportional hazard analysis using Model 1 (**a**) and Model 2 (**b**). Open and filled rhombus indicate reference, and significantly increased risk for each endpoint, respectively.

**Table 1 jcm-09-03963-t001:** Baseline and procedural characteristics of study patients categorized by gender and AF.

	Overall	FemaleAF (-)	Male AF (−)	FemaleAF (+)	Male AF (+)	*p*-Value
	*n* = 1088	*n* = 559, 51.4%	*n* = 266, 24.4%	*n* = 187, 17.2%	*n* = 76, 7.0%	
Age, years	84.0 ± 5.5	84.1 ± 5.2	83.0 ± 6.4	85.2 ± 4.9	84.2 ± 5.8	***p* < 0.001**
BMI, kg/m^2^	22.3 ± 3.7	22.4 ± 3.9	22.4 ± 3.0	22.1 ± 4.3	22.4 ± 3.1	*p* = 0.78
NYHA class, III or IV, n	561 (51.7%)	270 (48.3%)	133 (50.0%)	112 (59.9%)	46 (60.5%)	***p* = 0.018**
Logistic EuroSCORE, %	12.8 (9.5, 19.0)	12.8 (9.5, 17.7)	11.4 (7.6, 19.3)	16.1 (12.0, 22.4)	13.6 (8.7, 26.1)	***p* < 0.001**
EuroSCORE II, %	4.5 (2.8, 7.0)	4.5 (2.8, 6.3)	3.5 (2.3, 6.5)	5.3 (3.5, 8.5)	5.6 (2.5, 10.0)	***p* < 0.001**
STS-PROM score, %	5.7 (3.8, 8.3)	5.6 (3.9, 7.7)	4.9 (3.3, 7.3)	7.2 (5.4, 10.3)	6.0 (3.7, 9.3)	***p* < 0.001**
**Comorbidities**						
History of heart failure, n	320 (29.4%)	132 (23.6%)	65 (24.4%)	85 (45.5%)	38 (50.0%)	***p* < 0.001**
Hypertension, n	841 (77.3%)	445 (79.6%)	201 (75.6%)	140 (74.9%)	55 (72.4%)	*p* = 0.28
Diabetes mellitus, n	262 (24.1%)	117 (20.9%)	83 (31.2%)	41 (21.9%)	21 (27.6%)	***p* = 0.0097**
Cancer, n	207 (19.0%)	96 (17.2%)	59 (22.2%)	36 (19.3%)	16 (21.1%)	*p* = 0.37
History of stroke, n	124 (11.4%)	49 (8.8%)	35 (13.2%)	26 (13.9%)	14 (18.4%)	***p* = 0.023**
COPD, n	112 (10.4%)	41 (7.4%)	41 (15.6%)	22 (11.9%)	8 (10.5%)	***p* = 0.0037**
CKD (stage 3 or more), n	713 (65.5%)	353 (63.2%)	160 (60.2%)	144 (77.0%)	56 (73.7%)	***p* < 0.001**
PAD, n	179 (16.5%)	75 (13.4%)	53 (19.9%)	37 (19.8%)	14 (18.4%)	*p* = 0.052
OMI, n	65 (6.0%)	20 (3.6%)	31 (11.7%)	4 (2.1%)	10 (13.2%)	***p* < 0.001**
History of coronary revascularization *, n	240 (22.1%)	94 (16.8%)	97 (36.5%)	28 (15.0%)	21 (27.6%)	***p* < 0.001**
p-PTAV, n	35 (3.2%)	20 (3.6%)	6 (2.3%)	9 (4.8%)	0 (0.0%)	*p* = 0.17
**Laboratory data**						
NT-proBNP, pg/mL	1154 (479, 3098)	868 (386, 2510)	1025 (409, 2563)	2173 (1111, 5141)	1784 (951, 5114)	***p* < 0.001**
Creatinine, mg/dL	0.9 ± 0.4	0.8 ± 0.3	1.1 ± 0.5	0.9 ± 0.4	1.1 ± 0.3	***p* < 0.001**
eGFR, ml/min	54.2 ± 18.9	55.4 ± 19.0	56.0 ± 19.4	49.3 ± 18.3	50.8 ± 15.6	***p* < 0.001**
Hemoglobin, g/dL	11.6 ± 1.6	11.3 ± 1.4	12.1 ± 1.6	11.4 ± 1.5	12.1 ± 1.8	***p* < 0.001**
Albumin, g/dL	3.7 ± 0.4	3.8 ± 0.4	3.7 ± 0.4	3.7 ± 0.4	3.7 ± 0.5	***p* = 0.018**
**Echocardiographic findings**						
LVEF, %	60.7 ± 10.8	62.6 ± 9.7	58.8 ± 11.6	59.8 ± 10.5	55.0 ± 13.1	***p* < 0.001**
AVA, cm^2^	0.66 ± 0.17	0.66 ± 0.17	0.70 ± 0.16	0.62 ± 0.18	0.68 ± 0.20	***p* < 0.001**
Peak gradient, mmHg	89.2 ± 31.5	92.2 ± 31.3	89.0 ± 30.2	85.5 ± 33.0	77.0 ± 29.8	***p* < 0.001**
Mean gradient, mmHg	51.2 ± 19.1	53.0 ± 19.6	51.0 ± 16.8	48.8 ± 20.3	43.8 ± 17.9	***p* < 0.001**
AR ≥ moderate, n	56 (5.1%)	18 (3.2%)	20 (7.5%)	12 (6.4%)	6 (7.9%)	***p* = 0.029**
MR ≥ moderate, n	57 (5.2%)	21 (3.8%)	11 (4.1%)	18 (9.6%)	7 (9.2%)	***p* = 0.0045**
TR ≥ moderate, n	57 (5.2%)	8 (1.4%)	5 (1.9%)	33 (17.6%)	11 (14.5%)	***p* < 0.001**
**Medications**						
Beta-blockers	382 (35.1%)	160 (28.6%)	81 (30.5%)	99 (52.9%)	42 (55.3%)	***p* < 0.001**
ACEIs/ARBs	596 (54.8%)	320 (57.3%)	140 (52.6%)	90 (48.1%)	46 (60.5%)	*p* = 0.11
Statins	571 (52.5%)	286 (51.2%)	149 (56.0%)	94 (50.3%)	42 (55.3%)	*p* = 0.48
Diuretics	516 (47.4%)	227 (40.6%)	115 (43.2%)	117 (62.6%)	57 (75.0%)	***p* < 0.001**
Oral anticoagulants	261 (24.0%)	27 (4.8%)	15 (5.6%)	162 (86.6%)	57 (75.0%)	***p* < 0.001**
**Procedural variables**						
Procedure time, min	73 (60, 100)	74 (60, 102)	73 (60, 97)	72 (59, 100)	73 (58, 101)	*p* = 0.95
Fluoroscopy time, min	20 (16, 27)	21 (16, 27)	20 (16, 27)	20 (16, 26)	20 (17, 28)	*p* = 0.88
Contrast medium volume, ml	61 (45, 96)	63 (46, 98)	60 (45, 94)	61 (43, 95)	55 (42, 86)	*p* = 0.30
**Approach plan**						
Conscious sedation, n	637 (58.6%)	336 (60.1%)	157 (59.0%)	101 (54.0%)	43 (56.6%)	*p* = 0.51
Transfemoral approach, n	993 (91.3%)	517 (92.5%)	242 (91.0%)	166 (88.8%)	68 (89.5%)	*p* = 0.58
Valve size, mm	24.8 ± 2.3	24.1 ± 2.1	26.1 ± 2.3	24.2 ± 2.1	26.2 ± 2.3	***p* < 0.001**
**Valve type**						
Edwards SAPIEN-XT, n	171 (15.7%)	92 (16.5%)	33 (12.4%)	32 (17.1%)	14 (18.4%)	*p* = 0.38
Edwards SAPIEN3, n	543 (49.9%)	275 (49.2%)	150 (56.4%)	84 (44.9%)	34 (44.7%)	*p* = 0.052
Medtronic CoreValve, n	29 (2.7%)	14 (2.5%)	11 (4.1%)	3 (1.6%)	1 (1.3%)	*p* = 0.30
Medtronic Evolut R, n	164 (15.1%)	88 (15.7%)	27 (10.2%)	40 (21.4%)	9 (11.8%)	***p* = 0.0072**
Medtronic Evolut PRO, n	128 (11.8%)	66 (11.8%)	30 (11.3%)	17 (9.1%)	15 (19.7%)	*p* = 0.13
Boston Scientific LOTUS, n	11 (1.0%)	5 (0.9%)	3 (1.1%)	2 (1.1%)	1 (1.3%)	*p* = 0.98
Balloon expandable, n	714 (65.6%)	367 (65.7%)	183 (68.8%)	116 (62.0%)	48 (63.2%)	*p* = 0.45

BMI: body mass index, STS-PROM: Society of Thoracic Surgeons Predicted Risk of Mortality, COPD: chronic obstructive pulmonary disease, CKD: chronic kidney disease, PAD: peripheral artery disease, OMI: old myocardial infarction, PTAV: percutaneous transcatheter aortic valvuloplasty, NT-proBNP: N-terminal pro-brain natriuretic peptide, eGFR; estimated glomerular filtration rate, LVEF: left ventricular ejection fraction, AVA: aortic valve area, AR: aortic regurgitation, MR: mitral regurgitation, TR: tricuspid regurgitation, ACEis: angiotensin converting enzyme inhibitors, ARBs: angiotensin II receptor blockers. * status post percutaneous coronary intervention or coronary artery bypass graft. Bold values indicate statistical significance at the *p* < 0.05 level.

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
