# Peer review of "Simultaneous Estimation of Gender Male and Atrial Fibrillation as Risk Factors for Adverse Outcomes Following Transcatheter Aortic Valve Implantation"

_jcm, 2020, doi:10.3390/jcm9123963_

Round 1

Reviewer 1 Report

This is the study conducted by Shikata et al to assess the impact of gender with/without presence of AF onto cardiovascular outcomes among patients undergoing TAVR. The author has found that male gender with AF conveyed the highest risk of all-cause mortality/ CV death and the composite outcomes of HF hospitalization and all-cause death. Overall, this study is well-written and good design in retrospective fashion. However, I have a couple questions which need to be addressed 

Major

Methodology

  1. For AF in this study, how did authors define ?, please illustrate more
  2. Have authors attempted analyzing gender and the presence of AF independently? According to the current manuscript, authors combined them together more dependent. It would be great if we have the separate HR of each element independently (under univariate and multivariate model)
  3. If authors have analyzed as above, I recommend performing interaction test to determine if gender and sex interact synergistically or effect modification (one way the other)
  4. I would like to admire the authors for deliberately collecting relevant information for variates adjustment as of retrospective study limitation. Nevertheless, have author used frailty index to help adjust, I have noticed that average albumin in this study seems to be not different much but in elderly population, this is very important and has both clinical and statistical impacts on CV outcomes. (The Effect and Relationship of Frailty Indices on Survival After Transcatheter Aortic Valve Replacement. JACC Cardiovasc Interv 2020;13:219-231)
  5. How is disabling stroke defined in this study?
  6. This is optional but it would be also very helpful to have. Is there any information about pacemaker implantation rates, rates of transvenous  pacer use? I am wondering as this might be another confounding factor if patient has sick sinus syndrome as an underlying especially in AF patients, complicating the overall results in this study.

Discussion and limitation

  1. Once the authors resolve my most curious question (independency between male gender and presence of AF as narrated above), please discuss the relevance issues related to this
  2. Given the study design, authors should discuss retrospective design as one of the limitations in the current study

Reviewer 2 Report

I have read with interest the study entitled: “Simultaneous estimation of gender male and atrial fibrillation as risk factors for adverse outcomes following transcatheter aortic valve implantation”. In this paper authors try to address the combined impact of gender and AF history on TAVI outcomes. By a retrospective analysis of a multicenter TAVI registry, authors conclude that male gender and/or AF existence have a slightly additive prognostic effect, even though not synergistic, on cardiovascular mortality and HF hospitalization after TAVI.

First of all, authors should be praised for this very well written paper. Despite this fact, I have some serious concerns that should be addressed:

1)       The impact of gender and AF history on short- and long-term outcomes after TAVI has been extensively studied in the literature (even though not in combination always). Under these circumstances, what novel comes this study to add? Please comment.

2)       I totally agree that rather contradictory results exist about female gender, however, I can’ t see how a registry characterized by female predominance may persuade about the “objectivity” of conclusions reached. The very same problem exists with the very “limited” number of male patients with AF consisting the 4th group (only 76 patients). Despite all the precautionary statistical measures taken, such a small number of patients (compared to the rest of the groups) may “inflate” incidence rates. Additionally, sth to worry about is if the small number of male AF(+) may justify the large number of variables included in the models 1 and 2 used for the multiple regression analysis.

3)       The clinical relevance and importance of the conclusions reached is not clearly presented. Since AF history and/or gender are already included in STS risk score (both factors) and EUROSCORE II (gender) commonly used for risk stratification before TAVI, should authors (based on their data) suggest another “risk” score, better suited for TAVI patients? 

Reviewer 3 Report

The manuscript by Chikata et al., entitled “Simultaneous estimation of gender male and atrial fibrillation as risk factors for adverse outcomes following transcatheter aortic valve implantation ", aimed to evaluate the sex-specific impact of AF on outcomes after TAVI.

This is a large retrospective observational study that included more than 1000 patients divided into 4 groups, female and male without atrial fibrillation [F AF(-) and M AF(-)] and female and male with atrial fibrillation [F AF(+) and M AF(+)]. The results of this study suggested that being male and/or having AF leads to the worst outcome after TAVI.

Overall, the manuscript is well written and clear. It also addresses an important topic that is still under debate. However, I do have several concerns that should be addressed.

  1. It is not clear why the author compare the F AF(-) group reference group, with three other groups. Recent evidence indicated that females have fewer complications and better outcomes after TAVI. In addition, AF was recognized as an independent risk factor for morbidity and mortality. Did the authors compare M AF(-) vs M AF(+)? This comparison seems to be not significant and thus AF, in the male group, does not add any risk. Did the authors compare M AF(-) vs F AF(+)? Also here it looks like no difference is present. However, if this is the case, being F AF(+) has the same prognostic impact as being M AF(-). The authors correctly stated that there is no synergistic prognostic impact but multiple comparisons changing the reference to evaluate other potential differences should be performed and discussed.

  1. Probably, for the assessment of simultaneous estimation of two factors (male sex and AF) the area under the receiving operator characteristic (ROC) curve (AUC) with the measurement of accuracy, specificity, sensitivity, negative predictive value, and positive predictive value could be calculated.

  1. Five-years mortality for all causes, cardiovascular mortality, and mortality for all causes combined with heart failure re-hospitalization were evaluated as primary and secondary endpoints in this study. It is unclear why mortality for all causes and HF re-hospitalization were assessed in combination. It would be interesting to know what the HF re-hospitalization was in all studied groups after 5 years of follow-up, taking into account that approximately 50% of patients with AF and 20% of patients without AF already had a history of HF at the baseline.

  1. The number of covariates in the multivariate model is quite a lot. In model 1, the authors corrected the Cox regression for 15 variables + 4 groups with a total of 19 co-variates. The number of recorded events, less than 190, is barely sufficient for the composite of all-cause death and HF hospitalization. In addition, the patient’s characteristic table clearly shows that there are even more covariates that should be taken into account to support the author's hypothesis. I suggest performing a propensity score matching and then re-do the analysis to confirm the findings or if this is not possible create a propensity score variable and adjust the analysis for this one variable. For example, a similar approach has been used previously in an article published in this Journal (J Clin Med. 2019 Aug 5;8(8):1172. doi: 10.3390/jcm8081172. PMID: 31387275).

Round 2

Reviewer 1 Report

Thank you for authors response. All my suggestion have been well addressed

Reviewer 2 Report

I would like to thank authors for taking into consideration my comments and thoughts. I believe that the modifications applied in the manuscript have significantly improved its content.

Reviewer 3 Report

I thanks the Authors that have replied to my comments.

I now think that the manuscript is acceptable in this form.